# Asian Flush Gene Variant Enhances Cellular Immunogenicity of COVID-19 Vaccine: Prospective Observation in the Japanese General Population

**DOI:** 10.3390/vaccines12091015

**Published:** 2024-09-05

**Authors:** Sudarma Bogahawaththa, Megumi Hara, Takuma Furukawa, Chiharu Iwasaka, Takeshi Sawada, Goki Yamada, Mikiko Tokiya, Kyoko Kitagawa, Yasunobu Miyake, Mizuho Aoki Kido, Yoshio Hirota, Akiko Matsumoto

**Affiliations:** 1Laboratory of Biochemistry, Department of Applied Biochemistry and Food Science, Faculty of Agriculture, Saga University, Honjo, Saga 840-8502, Japan; sz1944@cc.saga-u.ac.jp; 2Department of Social and Environmental Medicine, Saga University, Nabeshima, Saga 840-8501, Japan; sx4932@cc.saga-u.ac.jp; 3Department of Preventive Medicine, Faculty of Medicine, Saga University, Nabeshima, Saga 840-8501, Japan; harameg@cc.saga-u.ac.jp (M.H.); sr0753@cc.saga-u.ac.jp (T.F.); sy7509@cc.saga-u.ac.jp (C.I.); 4Clinical Research Center, Saga University Hospital, Nabeshima, Saga 840-8501, Japan; 5Department of Physical Activity Research, National Institutes of Biomedical Innovation, Health and Nutrition, KENTO Innovation Park NK Building, 3–17, Senriokashinmachi, Settsu 566-0002, Japan; 6Department of Histology and Neuroanatomy, Faculty of Medicine, Saga University, Nabeshima, Saga 840-8501, Japan; 22624004@edu.cc.saga-u.ac.jp (T.S.); kido@cc.saga-u.ac.jp (M.A.K.); 7United Graduate School of Agriculture, Kagoshima University, Kagoshima 890-0065, Japan; 23975002@edu.cc.saga-u.ac.jp; 8Division of Ultrastructural Cell Biology, Department of Anatomy, Faculty of Medicine, University of Miyazaki, Miyazaki 889-1692, Japan; kyoko_kitagawa@med.miyazaki-u.ac.jp; 9Division of Molecular and Cellular Immunoscience, Department of Biomolecular Sciences, Faculty of Medicine, Saga University, 5-1-1 Nabeshima, Saga 849-8501, Japan; ymiyake@cc.saga-u.ac.jp; 10Clinical Epidemiology Research Center, SOUSEIKAI Medical Group (Medical Co., LTA), Fukuoka 813-0017, Japan; hiro8yoshi@lta-med.com

**Keywords:** ALDH2, rs671, COVID-19, enzyme-linked immunospot, cellular immunity, T cell

## Abstract

We previously reported a reduced humoral immune response to the COVID-19 vaccines. Subsequently, we observed a lower susceptibility to COVID-19 in individuals carrying the *ALDH2* rs671 variant through a web-based retrospective survey. Based on these findings, we hypothesized that rs671 variant was beneficial for cellular immunity against COVID-19. Using the IFN-γ enzyme-linked immunospot (ELISPOT) assay, we assessed cellular immunity before and after COVID-19 vaccination in two subcohorts of a previously reported cohort. Subcohort 1 (26 participants) had six repeated observations at baseline after one to three doses, whereas subcohort 2 (19 participants) had two observations before and after the third dose. ELISPOT counts at six months after the second dose increased from baseline in carriers of the rs671 variant but not in non-carriers. A positive effect of rs671 on ELISPOT counts was estimated using a mixed model (183 observations from 45 participants), including the random effect of subcohort, repeated measures, and fixed effects of vaccine type, age, sex, height, lifestyle, steroid use, and allergic disease. There was no association between ELISPOT counts and specific IgG levels, suggesting a limitation in estimating protective potential by humoral response. Our sequential observational studies suggest a beneficial effect of the rs671 variant in SARS-CoV-2 infection via enhanced cellular immune response, providing a potential basis for optimizing preventive measures and drug development.

## 1. Introduction

The COVID-19 pandemic was a public health emergency because of its widespread and rapid transmission. The virus spreads primarily through respiratory droplets released when an infected person coughs, sneezes, or talks. Symptoms of COVID-19 can range from mild respiratory problems to severe conditions such as pneumonia and acute respiratory distress syndrome [1]. Vaccination is essential to enhance adaptive immunity, which plays a key role in the management of the COVID-19 pandemic [2] by stimulating the immune system to mount a protective response, including both cellular and humoral immunity [3].

The rs671 variant, a known East Asian-specific genetic diversity, is a missense mutation in aldehyde dehydrogenase 2 (*ALDH2*). This mutation involves a substitution of glutamic acid (GAA) for lysine (AAA) at amino acid position 504 in the immature enzyme or position 487 in the mature enzyme known as Glu504Lys or Glu487Lys. This change disrupts the three-dimensional structure of the enzyme and affects its functionality [4]. ALDH2 plays a crucial role in ethanol metabolism by oxidizing acetaldehyde, a toxic product of ethanol oxidation, into the less harmful substance, acetate [5,6]. Individuals carrying the rs671 variant found almost exclusively in East Asia have substantially reduced ALDH2 enzymatic activity, leading to the accumulation of acetaldehyde in the body when they consume alcohol [7]. This accumulation causes vasodilation, resulting in skin flushing, increased heart rate, nausea, and other unpleasant symptoms [8], a phenomenon known as the Asian flush [9,10]. Many other phenotypes of the rs671 variant have been reported [11,12,13], including alcohol-irrelevant disease risks, such as vasospastic angina [14] and cognitive impairment [15].

Based on our previous findings on the association between rs671 and immune function (i.e., rs671 influences basal T-cell subpopulation [16] and the efficiency of the inhibitors of immune checkpoint [12]), we launched a prospective cohort study to investigate the immunogenicity of the COVID-19 mRNA vaccine and found an inverse association between the rs671 variant and IgG production [17]. Subsequently, a web-based retrospective cohort study revealed lower susceptibility to COVID-19 among rs671 variant carriers [18]. These seemingly contradictory findings suggest the involvement of cellular immunity, which plays a greater role than humoral immunity in protecting against COVID-19 [19], as shown by the successful recovery of patients with X-linked agammaglobulinemia and genetic deficiency of mature B lymphocytes from COVID-19 [20].

To test our hypothesis, we designed an additional study using frozen blood cells from the above vaccine immunogenicity study to assess immune cell counts specific for SARS-CoV-2 before and after COVID-19 vaccination.

## 2. Materials and Methods

This study was approved by the Ethics Committee, School of Medicine, Saga University, Saga, Japan (No. R2-44 and R3-9). Written informed consent was obtained from all participants before any study procedure was performed.

### 2.1. Design of This Study and Participants

This study used a database and frozen blood cells from participants in our previously reported cohort [17]. To maximize statistical power with limited financial resources, rs671 minor allele carriers were prioritized over major allele carriers. The participants were employees and students at Saga University, Japan, aged 20 years or older, who voluntarily received the COVID-19 vaccine administered at Saga University. Pregnant women and those who had been infected or suspected of being infected with COVID-19 were excluded from this study.

As shown in Figure 1, university employees (*n* = 16) and some of the students (*n* = 10) completed two doses of mRNA-1273 (Moderna Inc., Cambridge, MA, USA/Takeda Pharmaceutical Co., Ltd., Tokyo, Japan) (100 µg) starting 9–11 August 2021, and a booster dose of BNT 162b2 (Pfizer Inc., New York, NY, USA/BioNTech SE, Mainz, Germany) (30 µg) (subcohort-1), and the rest of university students (*n* = 19) received three doses of BNT 162b2 (Pfizer) starting 19 May 2021 (subcohort-2). Vaccination history was recorded using a vaccination certificate issued by the local government of Saga Prefecture, Japan. mRNA-1273 and BNT 162b2 were completed at an interval of three or four weeks, and booster vaccination with BNT 162b2 was given six or eight months after the second dose for subcohorts 1 and 2, respectively. Blood samples were collected at baseline, three weeks after the first dose, one, three, and six months after the second dose (two, four, and seven months after the first dose), and one month after the third dose (8 months after the first dose) for subcohort 1. For subcohort 2, blood samples were collected eight months after the second dose and one month after the second dose. Blood sample collection for subcohort 2 began before the booster dose because of a change in the budget plan.

### 2.2. Questionnaire

A self-administered questionnaire was used to obtain participants’ general information, as previously reported [17]. Briefly, sex, date of birth, disease (chronic lung disease, heart disease, stroke, kidney disease, liver disease, blood disease, diabetes, immunodeficiency, malignancy, collagen disease, and allergic disease), treatment history, cigarette smoking, alcohol drinking, and exercise habits were recorded. Ethanol intake was assessed and categorized into <1, <20, and ≥20 g/day over the previous 6 months, normalized to a 60 kg body weight. Exercise habit was chosen from no habit, <1 day/week, 1 to 3 days/week, and ≥3 days/week. The question, “Do you feel psychological stress?” was asked to evaluate perceived stress on a five-point scale: no (0); mostly no (1); unsure (2); quite often (3); and yes (4).

### 2.3. Genotyping

The *ALDH2* rs671 genotype was determined as described previously [17]. Briefly, blood clots (0.1 mL) were treated with 0.4 mL proteinase K solution (proteinase K at 1–10 U/mL in 0.01 M Tris-HCl, pH 8 with 0.01 M EDTA and 0.5% sodium dodecyl sulfate) overnight at 56 °C. After adding 0.5 mL of TE-saturated phenol and mixing, samples were incubated on ice, centrifuged, and the aqueous layer collected. An amount of 95% ethanol (0.5 mL) was added, mixed and incubated. DNA was precipitated, washed with 70% ethanol, and dissolved in 100 µL of DNase-free water. Extracted DNA was analyzed using a TaqMan^®^ SNP genotyping assay system (Thermo Fisher Scientific, Waltham, MA, USA).

### 2.4. Separation of Peripheral Blood Mononuclear Cells (PBMCs)

Peripheral blood was collected in EDTA-2K vacuum blood sampling tubes. To collect PMBCs, blood samples were subjected to gradient separation with Lymphoprep™ (Serumwerk Bernburg, Burnburg, Germany) within the day of sampling, according to the manufacturer’s instruction. The cell suspension was extracted at the border of the top, and the second layer was collected, washed with >5 times the volume of saline solution, and dissolved in KM Banker II (Kohjin Bio Co., Ltd., Saitama, Japan), a culture medium suitable for the long-term cryopreservation of lymphocytes. A small portion of the PBMC suspension was subjected to cell counting after 0.4 *w*/*v*% trypan blue. Finally, the PBMC samples were gently cooled and preserved at −80 °C until the enzyme-linked immunospot (ELISPOT) assay.

### 2.5. ELISPOT Assay

The ELISPOT assay was used to assess the functional activity of T cells by measuring interferon gamma (IFN-γ) responses in PBMCs. This analysis provides insight into the cellular immune response induced by vaccination schedule. A human IFN-γ ELISPOT kit (Mabtech AB, Nacka Strand, Sweden) was used to evaluate cellular immunity against SARS-CoV-2 using lyophilized 315-peptide mixtures of SARS-CoV-2 spike glycoprotein, PepMixTM SARS-CoV-2 (JPT Peptide Technologies, Berlin, Germany), the original strain (PM-WCPV-S-2) as antigens. The peptide mixture was dissolved in DMSO at a concentration of 0.5 ug/peptide, and a DMSO addition group was provided as a negative control. Similarly, a nuclear factor kappa B activator, phorbol myristate acetate, and ionomycin addition group were provided as positive controls. The assay was performed following the manufacturer’s instructions using an ImmunoSpot S6 VERSA spot counter (Cellular Technology Limited, Shaker Heights, OH, USA).

### 2.6. Cross-Reactivity with SARS-CoV-2 Delta and Omicron Variants

Assessing vaccine efficacy across SARS-CoV-2 variants is essential to ensure that vaccines continue to protect against viral changes. Samples with sufficient cell numbers for this analysis were evaluated for reactivity against two different peptide mixtures: the original and delta variants (PM-SARS2-SMUT06-1) and the original and omicron variants (PM-SARS2-SMUT08-1). Cross-reactivity was evaluated between the original and delta variants in PBMCs from 74 blood samples from subcohort 1 (three weeks after the first dose and one and three months after the second dose) and subcohort 2 and between the original and omicron variants in PBMCs from 55 samples from subcohort 1 (six months after the second dose and one month after the third dose).

### 2.7. Statistical Analysis

SARS-CoV-2 infection during follow-up was detected by an increase in anti-SARS-CoV-2 S1 protein IgN levels or specific IgG and T cell counts without vaccination, and the corresponding observation points were removed after the suspected infection. Spearman’s rank correlation ρ was calculated to examine the non-linear associations between rank or continuous variables, such as rs671 variant number, ethanol intake, ELISPOT counts, and IgG levels. Mixed models were employed to analyze the relationship between the number of rs671 variant alleles using a total of 183 observations from 45 participants and the log-transformed levels of specific T cells, allowing for repeated measures and the random effect of different subpopulations (proc mixed by SAS9.4 TS Level 1M5 for Windows, SAS Institute, Cary, NC, USA). ELISPOT counts were converted into log values to approximate a normal distribution. Statistical significance was set at *p* < 0.05.

## 3. Results

### 3.1. Baseline Characters of the Participants

Table 1 shows the baseline characteristics of each subcohort, including 30 males and 15 females with confirmed rs671 genotypes: wild-type homozygous (GG type, *n* = 20); heterozygous (GA type, *n* = 17); and variant homozygous (AA type, *n* = 8). The variant allele carriers (GA and AA) showed lower daily ethanol intake (g/day) compared to the GG group (ρ = −0.42, *p* < 0.01). There were no significant variations in exercise habits, perceived stress, or prevalence of allergic diseases among the groups (*p* > 0.4, Fisher’s exact test). Steroid use was reported by only two individuals in the GG group.

### 3.2. Changes in Cellular Immune Response after Vaccination

As depicted in Figure 1, we conducted an IFN-γ ELISPOT assay to quantify the number of T cells producing IFN-γ in response to SARS-CoV-2 specific antigens. The variant carriers tended to have higher counts than wild-type carriers at most time points. For example, at baseline in subcohort 1, wild and variant types had median counts of 0.5 and 5.5 spot-forming cells per 2 × 10^5^ PBMC, respectively. One month after the second dose, the medians reached 139.5 and 196, respectively. Six months after the second dose, the number decreased to 36 in the wild-type group, showing an indistinguishable level from baseline (*p* = 0.10, Wilcoxon rank-sum test), whereas 116 in the variant-type group were higher than baseline (*p* < 0.01). This effect could not be analyzed in subcohort 2 due to lack of samples. In subcohort 1, it was also possible to evaluate the booster effect by comparing one month after the second dose with one month after the third dose. As a result, the booster effect tended to be observed only in variant carriers (*p* = 1.00 and 0.13 for the wild-type and variant-type, respectively, Wilcoxon rank-sum test).

As shown in Table 2, the effects of rs671 were estimated using multivariate mixed models with repeated measures. Model 1, including categorical time points, vaccine type, age, sex, and the number of variant alleles, estimated a positive effect of the variant (β = 0.27 per allele, *p* = 0.01). Model 2, additionally including lifestyle, perceived stress, steroid use, and allergic disease history, gave a similar estimation (β = 0.3, *p* = 0.007).

As reported previously, anti-SARS-CoV-2 spike 1 IgG (anti-S1 IgG) and the rs671 variant were inversely associated in the current cohort [17]. Therefore, humoral and cellular immunity were correlated with rs671 in the opposite direction. As shown in Figure 2, no rank correlation was observed between specific IgG and T-cell responses obtained from the same blood samples. In a mixed model to estimate log-transformed specific T-cell count by categorical time points and log-transformed anti-S1 IgG considering repeated measures and subcohorts, anti-S1 IgG was not a predictor of specific T-cell count (partial correlation coefficient = −0.05, *p* = 0.7, 183 observations).

## 4. Discussion

Confirming our hypothesis, we found for the first time a positive association between the rs671 variant and cellular immunity characterized by SARS-CoV-2 spike protein-specific IFN-γ+ T-cell count after COVID-19 vaccination in a prospective observation of the general population in Japan. Cellular immunogenicity remained detectable six months after the second dose only in the rs671 variant group. The humoral immune reaction measured using anti-S1 IgG was not associated with the cellular immunity observed in the current study.

Although ALDH2 is known as an alcohol-metabolizing enzyme, its essential role is in the metabolism of endogenous aldehydes, such as formaldehyde and 4-hydroxy-2-nonenal (4-HNE), which are produced in the body as byproducts of normal cellular processes, such as spontaneous generation during one-carbon metabolism [21], demethylation of sarcosine [22,23] for formaldehyde, and peroxidation of arachidonic acid [24] for 4-HNE. The accumulation of these aldehydes due to low ALDH2 activity is a potential pathogen via adduct formation with DNA [25,26] and proteins [27], resulting in carcinogenesis and functional alterations of the molecules [28,29].

Endogenous formaldehyde and 4-HNE inhibit the mechanistic target of rapamycin (mTOR) signaling by modulating mTOR complex components [30]. The mTOR protein kinase serves as a central regulator of cell growth, proliferation, metabolism, and survival. It exists in two distinct multiprotein complexes: mTOR complex 1 (mTORC1) and mTOR complex 2 (mTORC2). mTORC1 primarily regulates the processes related to cell growth, protein synthesis, and metabolism in response to nutrients, growth factors, and energy status. Inhibition of mTORC1 signaling by formaldehyde and 4-HNE may have implications for the maintenance of memory T cells, which are a small population of long-lived immune cells derived from effector T cells that provide rapid and specific responses upon re-exposure to previously encountered pathogens or antigens [31]. In our observation, the variant type showed higher T-cell counts than the wild type; however, multiple testing of rank correlation between the variant allele number and specific T-cell count for each time point gave the only significance at six months after the second dose (ρ = 0.5, *p* = 0.02). These findings suggest that the accumulation of endogenous aldehydes may inhibit mTOR signaling, thereby promoting memory T cells.

Another possible mechanism is the inhibition of T-cell glycolysis by endogenous aldehydes, as observed in acetaldehyde-exposed T cells [32]. By interfering with glycolytic activity, endogenous formaldehyde, and 4-HNE may shift cellular metabolism towards alternative pathways such as fatty acid oxidation, the pentose phosphate pathway, and oxidative phosphorylation for energy production. Memory T cells favor these alternative pathways, which are crucial for providing energy and biosynthetic precursors, maintaining the redox balance, ensuring the metabolic fitness, longevity, and functionality of memory T cells. We previously performed lymphocyte subset analysis in the general Japanese population and found that rs671 variant carriers were associated with lower CD4+ and CD8+ T-cell counts [16]. Combined with the present findings, rs671 variant carriers may have a higher rate of memory T-cell counts, even though the total number of T cells is lower.

Although the COVID-19 vaccines in our study were designed for the original strain, PBMCs in the current study reacted with original, delta, and omicron variants (R squared = 0.71 and 0.82 for original vs. delta and original vs. omicron, respectively, Figure 3). In contrast, humoral immunity is expected to escape, particularly for omicron variants [33,34]. Given this evidence, the hypothetical benefit of the rs671 variant on memory T-cell survival may persist throughout the pandemic; the vaccine designed for the original strain or history of exposure to a similar virus may induce cellular immunity more efficiently in rs671 variant carriers, and its protective effect may be effective for the virus variants that show immune evasion of humoral immunity. The lower incidence of COVID-19 and related hospitalizations in our previous report [18] might, thus, be explained by the current findings; however, the reported strong effect in the early phase (hazard ratio 0.2) may not be fully explained, suggesting the presence of additional defense mechanisms, such as antimicrobial effects of metabolic intermediates aldehydes [35]. For instance, we propose that the accumulation of formaldehyde in individuals with the rs671 variant may protect against COVID-19 through its bacteriostatic effect at the physiological levels (100 µM) [36]. This infection defense phenotype of the rs671 variant is a possible reason for its spread in East Asia following major lifestyle changes associated with the infectious hazards of rice cultivation [37,38,39].

This study has far-reaching implications. For example, the positive correlation between the rs671 variant and enhanced cellular immune response suggests an unrecognized mechanism of immune regulation that may lead to ideas for new drug and vaccine discovery. It also suggests the importance of personalized vaccine strategies that take into account not only genetic information but also the factors that may influence ALDH2 activity; for example, alcohol consumption [40] and non-steroidal anti-inflammatory drugs [41,42,43] may affect vaccine efficacy in part through reduced ALDH2 activity.

On the other hand, the interpretation of the results must be in light of the limitations. To address the limitation of the validity owing to the small sample size, we attempted to maximize the detection power using statistical modeling to obtain all observation points. It is impossible to expand the study size now that most of the population has had the opportunity to receive the COVID-19 vaccine or to be exposed to the SARS-CoV-2 virus. Validation experiments, such as cellular immune stimulation experiments using rs671 model animals, will be a reasonable option to gain scientific confidence. Targeting single cytokine production, IFN-γ, is another limitation because multi-cytokine evaluation gives us background information; interleukin 4 (IL-4), for example, is an indicator of antibody production systems [44]. In addition, the use of PBMC without subpopulation sorting limits the specificity of findings because several types of immune cells can produce IFN-γ, including CD4+ T helper cells, CD8+ cytotoxic T cells, and natural killer T cells. To overcome such limitations, the isolation of specific immune cell subpopulations using techniques such as flow cytometry will increase the specificity of the findings by focusing on individual cell types. Furthermore, evaluation of multiple cytokines, including IL-4 in addition to IFN-γ, will provide a more comprehensive understanding of immune responses.

## 5. Conclusions

This study revealed for the first time a positive association between the rs671 variant and enhanced cellular immunity following COVID-19 vaccination. Reduced ALDH2 activity may favor the development and maintenance of COVID-19-specific memory T cells. The current findings support the previously reported lower susceptibility to COVID-19 infection in rs671 variant carriers. These findings may help in designing effective vaccination strategies; however, the underlying mechanisms require further investigation.

## Figures and Tables

**Figure 1 vaccines-12-01015-f001:**
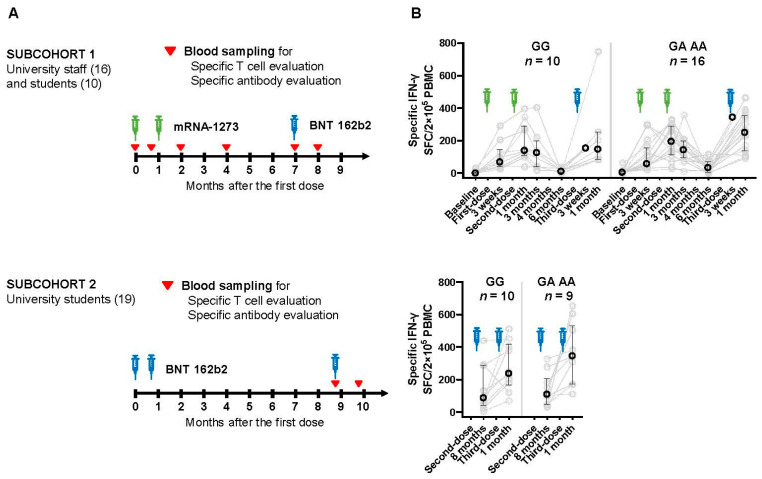
SARS-CoV-2 S1 protein-specific IFN-γ^+^ cell counts after COVID-19 vaccination. (**A**) Study design. Subcohort 1 included 26 participants who completed two doses of mRNA-1273 (100 µg) and a booster dose of BNT162b2 (30 µg). Subcohort 2 included 19 male students who received three doses of BNT162b2. The green and blue syringe icons indicate mRNA-1273 and BNT 162b2, respectively. Blood samples were collected as indicated in red triangles. (**B**) SARS-CoV-2 S1 protein specific IFN-γ+ cell counts. IFN-γ ELISPOT counts after immunological activation with 315-peptide mixtures of SARS-CoV-2 spike glycoprotein of the original strain were plotted. Grey circles indicate the individual ELISPOT counts. Black circles indicate the median ELISPOT count for each time point, with error bars representing the interquartile range.

**Figure 2 vaccines-12-01015-f002:**
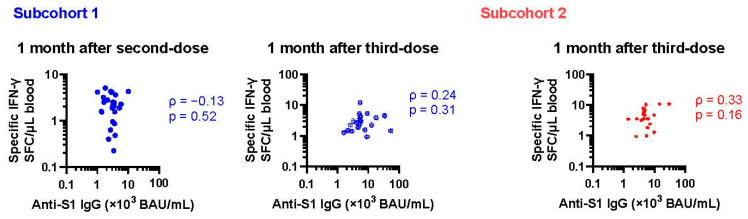
Association between humoral and cellular immunity. Anti-SARS-CoV-2 S1 IgG and IFN-γ+ PBMC count per 1 µL whole blood are plotted for 1 month after the second and third dose of the COVID-19 vaccine (blue and red dots represent subcohort 1 and 2, respectively). SFC: spot-forming cells. PBMC, peripheral blood mononuclear cells. Spearman’s rank correlation coefficients (ρ) and *p*-values are presented.

**Figure 3 vaccines-12-01015-f003:**
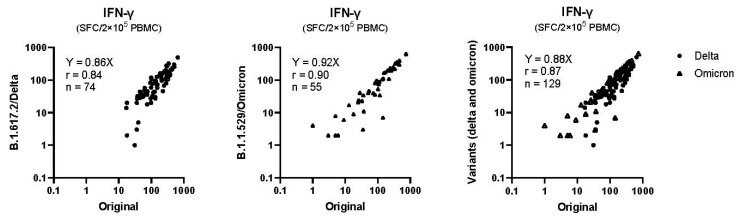
Cross-reactivity to SARS-CoV-2 delta and omicron variants. Cross-reactivity was assessed between the original and delta variants in PBMCs from 74 blood samples from subcohort 1 (three weeks after the first dose and one and three months after the second dose), subcohort 2, and between the original and omicron variants in PBMCs from 55 samples from subcohort 1 (six months after the second dose and one month after the third dose). PBMC, peripheral blood mononuclear cells.

**Table 1 vaccines-12-01015-t001:** Baseline characteristics of participants by *ALDH2* rs671 genotypes.

	Subcohort 1 University Employees and Students			Subcohort 2 University Students		
*n*	26			19		
Type of Vaccine	mRNA-1273 × 2 BNT162b2 × 1			BNT162b2 × 3		
*ALDH2* rs671	GG	GA	AA	GG	GA	AA
Males, *n*	3	7	1	10	4	5
Females, *n*	7	6	2	0	0	0
Age, years						
Median	39	47	21	23	23	22
(IQR)	(22–56)	(21–55)	(21–42)	(22–23)	(23–24)	(22–23)
Body height, cm						
Median	162	166	157	171	173	172
(IQR)	(155–166)	(163–170)	(151–163)	(167–175)	(167–178)	(170–173)
Cigarette smoke, yes	0	1	0	0	0	0
Ethanol intake *						
<1 g/d	6 (60%)	9 (69%)	3 (100%)	2 (20%)	2 (50%)	4 (80%)
≥1, <20 g/d	4 (40%)	2 (15%)	0 (0%)	8 (80%)	2 (50%)	1 (20%)
≥20 g/d	0 (0%)	2 (15%)	0 (0%)	0 (0%)	0 (0%)	0 (0%)
Exercise habit						
No habit	3 (30%)	4 (31%)	1 (33%)	1 (10%)	1 (25%)	2 (40%)
<1 d/w	2 (20%)	2 (15%)	0 (0%)	4 (40%)	1 (25%)	1 (20%)
1 to 3 d/w	2 (20%)	4 (31%)	1 (33%)	2 (20%)	2 (50%)	2 (40%)
≥3 d/w	3 (30%)	3 (23%)	1 (33%)	3 (30%)	0 (0%)	0 (0%)
Perceived stress						
0 (no)	1 (10%)	5 (38%)	1 (33%)	5 (50%)	1 (25%)	1 (20%)
1	3 (30%)	1 (8%)	0 (0%)	1 (10%)	1 (25%)	0 (0%)
2	2 (20%)	2 (15%)	0 (0%)	1 (10%)	2 (50%)	3 (60%)
3	4 (40%)	4 (31%)	0 (0%)	3 (30%)	0 (0%)	1 (20%)
4 (yes)	0 (0%)	1 (8%)	2 (67%)	0 (0%)	0 (0%)	0 (0%)
Steroid use, yes	0 (0%)	0 (0%)	0 (0%)	2 (20%)	0 (0%)	0 (0%)
Allergic disease, yes	2 (20%)	6 (46%)	1 (33%)	5 (50%)	1 (25%)	1 (20%)

The genotypes GG, GA, and AA correspond to *ALDH2*1/*1*, *ALDH2*1*/**2*, and *ALDH2*2/*2*, respectively. IQR stands for interquartile range. * Ethanol intake was adjusted for body weight (g/day/60 kg body weight). Due to financial limitations, we selected 19 male subjects to include all homozygous variant allele (AA) carriers among the 42 participants in subcohort 2.

**Table 2 vaccines-12-01015-t002:** Estimated effects of baseline characteristics on log-transformed IFN-γ ELISPOT count.

	AIC = 486		AIC = 498	
	183 Datapoints (*n* = 45)		183 Datapoints (*n* = 45)	
Fixed Effect	β	*p*-Value	β	*p*-Value
BNT162b2 (reference)				
mRNA-1273	−0.58	0.067	−0.64	0.065
Age (per 10 years)	0.08	0.108	0.09	0.137
Female sex	0.10	0.536	0.24	0.358
Height (per 10 cm)			0.04	0.802
Tobacco smoking, yes			−0.52	0.294
Ethanol intake (per category)			0.10	0.500
Exercise (per category)			0.07	0.337
Perceived stress (per category)			0.02	0.767
Steroid use, yes			0.17	0.759
Allergic diseases			−0.22	0.185
Number of rs671 variant allele	0.27	0.010	0.30	0.007

β, Partial correlation coefficient. All models incorporate the fixed effects of post-vaccination timing as a categorical variable alongside the variables listed in the table.

## Data Availability

The datasets generated and/or analyzed in the current study are not publicly available to protect the privacy of the participants; however, they are available from the corresponding author upon reasonable request.

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
