# Peer review of "Asian Flush Gene Variant Enhances Cellular Immunogenicity of COVID-19 Vaccine: Prospective Observation in the Japanese General Population"

_vaccines, 2024, doi:10.3390/vaccines12091015_

Round 1
Reviewer 1 Report
Comments and Suggestions for Authors
Manuscript “Asian Flush Gene Variant Enhances Cellular Immunogenicity of COVID-19 Vaccine: Prospective Study in the Japanese General Population” by Sudarma and colleague based on their previous study further to explored the effect of rs671 variants on cellular immunity of COVID-19, however, there are some consideration need to be resolved.
1) The study should describe how many specific IFN-γ+ cell counts to represent clinical significance. The study showed six month after the second dose the wild-type group decreased to 36, and the variant-type only decreased to 116 which were statistically higher than baseline. In practical application , could 36 vs 116 SFC indicate more survival of T memory cells? And how the change of specific IFN-γ+ cell counts in the subcohort 2 after the second dose?It was lack of relevant description.
2) the study showed a positive effect of the variant using multivariate mixed models with repeated measure. The positive effect estimated is based on subcohort 1 or 2 or both? Please provide more details.
3) The authors had found that reduced T cell immunity in variant ALDH2 allele carriers. In current study, rs671 variants was likely to enhanced IFN-γ+ T cell production. How to explain seemingly contradictory findings? Pleased add it into discussion.
4) Discussion, the study only used singe cytokine produced by PBMC to explain the effect of the variant on cellular immunogenicity. The data is too simple to draw conclusive opinions. The relevant sentences is needed be revised. For example, Line 230-232 and Line 245-249.
5) In light of the limitation in the study, it is recommended to provide available solutions or next experiment plan.
Comments on the Quality of English Language
Minor editing of English language is required.
Author Response
Comment 1-1: The study should describe how many specific IFN-γ+ cell counts to represent clinical significance. The study showed six month after the second dose the wild-type group decreased to 36, and the variant-type only decreased to 116 which were statistically higher than baseline. In practical application , could 36 vs 116 SFC indicate more survival of T memory cells?
Reply: Thank you for your insightful comments. Based on current data and available research findings, it is challenging to determine a clinically significant level for ELISPOT counts, because IFN-γ cell count could vary depending on the methodology we used, peptide mixture or antigen types, amount of PBMC and the spot count reading device. In addition, The significance of ELISPOT counts can vary depending on the specific immune cell subpopulations being analyzed. For instance, the clinical relevance of responses from memory T cells versus effector T cells can differ substantially. Because our study did not analyze these specific subpopulations separately, it limits our ability to draw precise conclusions about clinical significant level of ELISPOT count. In our next experiment, we plan to analyse specific immune cell subsets and their responses following COVID-19 vaccination. By studying these subsets, such as memory T cells, effector T cells and other relevant populations, we aim to identify clinically relevant thresholds for their immune responses. This approach will help us to better understand how different subsets contribute to vaccine-induced immunity and could lead to more targeted strategies to improve vaccine efficacy.
We added a description regarding this limitation and next plan to the discussion part:
Line 323-327:
To overcome such limitations, isolation of specific immune cell subpopulations using techniques such as flow cytometry will increase the specificity of the findings by focusing on individual cell types. Furthermore, evaluation multiple cytokines, including IL-4 in addition to IFN-γ, will provide a more comprehensive understanding of immune responses.
Comment 1-2: And how the change of specific IFN-γ+ cell counts in the subcohort 2 after the second dose?It was lack of relevant description.
Reply: In subcohort 2, we only had samples taken before and after the third vaccination, so we could only observe the obvious result that ELISPOT counts increase after vaccination. However, since the data from subcohort 2 are useful for examining differences due to rs671, the observations were used in the mixed model. The mixed models included the 183 observations from 45 individuals from subcohorts 1 and 2. We have included this information in Table 2. We added a description of the above explanation.
Line 198 (result):
This effect could not be analyzed in subcohort 2 due to lack of sample.
Line 30 (abstract):
A positive effect of rs671 on ELISPOT counts was estimated using a mixed model (183 observa-tions from 45 participants), including the random effect of subcohort, repeated measures, and fixed effects of vaccine type, age, sex, height, lifestyle, steroid use, and allergic disease.
Comment 2: the study showed a positive effect of the variant using multivariate mixed models with repeated measure. The positive effect estimated is based on subcohort 1 or 2 or both? Please provide more details.
Reply: Thank you for your inquiry regarding the analysis of our study. We appreciate the opportunity to clarify our findings. We performed a repeated measures mixed model analysis based on data from both sub-cohorts. The mixed model approach allowed us to include multiple cohorts in a single model by using random intercepts. This method accounts for the variability between cohorts and helps control for individual differences, providing more accurate estimates of the effects across all participants.
Line 30 (abstract):
A positive effect of rs671 on ELISPOT counts was estimated using a mixed model (183 observa-tions from 45 participants), including the random effect of subcohort, repeated measures, and fixed effects of vaccine type, age, sex, height, lifestyle, steroid use, and allergic disease.
Line 170-173 (method):
Mixed models were employed to analyze the relationship between the number of rs671 variant alleles using a total of 183 observations from 45 participants and the log-transformed levels of specific T cells, allowing for repeated measures and the random effect of different subpopulations (proc mixed by SAS9.4 TS Level 1M5 for Windows, SAS Institute, Cary, NC, USA).
Comment 3: The authors had found that reduced T cell immunity in variant ALDH2 allele carriers. In current study, rs671 variants was likely to enhanced IFN-γ+ T cell production. How to explain seemingly contradictory findings? Pleased add it into discussion.
Reply: Thank you for raising this point. We previously performed lymphocyte subset analysis in the general Japanese population and found that rs671 variant carriers were associated with lower CD4+ and CD8+ T cell counts. Taken together with the present findings, rs671 variant carriers may have a higher rate of memory T cells, although the total number of T cells is lower. We have added the description to the discussion.
In discussion
Line 274-278:
We previously performed lymphocyte subset analysis in the general Japanese population and found that rs671 variant carriers were associated with lower CD4+ and CD8+ T cell counts [16]. Combined with the present findings, rs671 variant carriers may have a higher rate of memory T cell counts, even though the total number of T cells is lower.
Comment 4: Discussion, the study only used singe cytokine produced by PBMC to explain the effect of the variant on cellular immunogenicity. The data is too simple to draw conclusive opinions. The relevant sentences is needed be revised. For example, Line 230-232 and Line 245-249.
Reply: Thank you for your suggestion. This is one of the most important limitations of our study. We added the discussion for the limitation due to the single cytokine evaluation.
Line 321-326:
Targeting single cytokine production, IFN-γ, is another limitation, because mul-ti-cytokine evaluation gives us background information, IL-4 for example, an indicator of antibody production systems [45]. In addition, the use of PBMC without subpopulation sorting limits the specificity of findings because several types of immune cells can produce IFN-γ, including CD4+ T helper cells, CD8+ cytotoxic T cells, and natural killer T cells. To overcome such limitations, isolating specific immune cell subpopulations using techniques such as flow cytometry will enhance the specificity of the findings by focusing on individual cell types. Furthermore, evaluating multiple cytokines, including IL-4 in addition to IFN-γ, will provide a more comprehensive understanding of immune responses.
We also revised the suggested sentences to be less judgmental:
(before)
This finding is consistent with the hypothesis that accumulation of endogenous aldehydes inhibits mTOR signaling, providing acquired immunity via survival of memory T cells.
(after)
Line 263-265 (discussion):
These findings suggest a possibility that the accumulation of endogenous aldehydes may inhibit mTOR signaling.
(before)
Given this evidence, our hypothetical benefit of the rs671 variant on the survival of memory T cells may hold throughout the pandemic; the rs671 variant may promote COVID-19 protection via cellular immunity induced by the vaccine or exposure history to similar virus. The lower occurrence of COVID-19 and related hospitalizations in our previous report [22] is thus explainable by the current findings;
(after)
Line 283-297 (discussion):
Given this evidence, the hypothetical benefit of the rs671 variant on memory T-cell survival may persist throughout the pandemic; the vaccine designed for the original strain or history of exposure to a similar virus may induce cellular immunity more efficiently in rs671 variant carriers, and its protective effect may be effective for the virus variants that show immune evasion of humoral immunity. The lower incidence of COVID-19 and related hospitalizations in our previous report [18] might thus be explained by the current findings
Comment 5: In light of the limitation in the study, it is recommended to provide available solutions or next experiment plan.
Reply: Thank you for highlighting the importance of addressing the study's limitations. We have now included a section discussing potential solutions and future experimental plans to address these limitations. This addition provides a clearer pathway for further research and highlights the steps we intend to take to enhance the study's robustness and address the identified constraints.
Line 326-330 (discussion):
It is impossible to expand the study size now that most of the population has had the opportunity to receive the COVID-19 vaccine or to be exposed to the SARS-CoV-2 virus. Validation experiments, such as cellular immune stimulation experiments using rs671 model animals, will be a reasonable option to gain scientific confidence.
Reviewer 2 Report
Comments and Suggestions for Authors
Thank you for the nice research work.

Author Response
Review report entitled “Asian Flush Gene Variant Enhances Cellular Immunogenicity of COVID-19 Vaccine: Prospective Study in the Japanese General Population “submitted by Bogahawaththa et al., to Vaccine 2024 Covid-19 vaccination has a significant global health benefits and remains a reliable preventive measure to fight against disease. The present projet of Bogahawaththa and coauthors investigated the beneficial role and clinical relevance of the rs671 for COVID-19 in a Japanese population. While the study is limited to a small size, the authors demonstrated that carriers of this variant are less susceptible to the disease and are good responders to vaccination. By genotyping the ALDH2 gene and using ELISPOT counts in a subcohort population, they demonstrated beneficial effect of rs671 mutations after COVID-vaccination. All article sections are clear, informative and well writing. The methodology the authors used is very adequate to the project and results are nicely presented. The discussion section is very informative and supports the hypothesis the authors have suggested. In addition, the authors addressed their limits and gave perspectives.
Response:
Thank you very much for your positive comments.
Reviewer 3 Report
Comments and Suggestions for Authors
This is an interesting, well-written study based on the authors’ observation of a reduced humoral immune response to the COVID-19 vaccine and a subsequently discovered lower susceptibility to COVID-19 infection in individuals carrying the ALDH2 rs671 variant. The study is an extension of a previous prospective cohort study published in 2022 (ref 21) which investigated the immunogenicity of the COVID‐19 mRNA vaccine and found an inverse association between the rs671 variant and IgG production. The authors hypothesized that rs671 was beneficial for cellular immunity against COVID-19. The study used the IFN-γ enzyme-linked immunospot (ELISPOT) assay to evaluate cellular immunity before and after COVID-19 vaccination in two subcohorts of university students. Subcohort 1 (26 participants) had six repeated observations at baseline and after one to three doses, whereas sub-cohort 2 (19 participants) had two observations, before and after the third dose. ELISPOT counts at six months after the second dose increased from baseline in carriers of the rs671 variant, but not in non-carriers. A positive effect of rs671 on ELISPOT counts was estimated using a mixed model, including the random effect of subcohort, repeated measures, and fixed effects of vaccine type, age, sex, height, lifestyle, steroid use, and allergic disease. There was no association between ELISPOT counts and specific IgG levels, suggesting a limitation in estimating protective potential by humoral response. The study showed a lower humoral and higher cellular immunity against SARS-CoV-2 in rs671 variant carriers than non-carriers, suggesting a potential basis for optimizing preventive measures and possibly drug discovery.
I noted that an institutional ethics committee has not endorsed the study. This is necessary under such research designs but especially considering genotyping of individuals was part of the methodology.
The supplementary data commentary on cross-reactivity to SARS-CoV-2 delta and omicron variants was not completed. Readers might like to read the commentary but also see the figure, given it adds interest to the paper.
Author Response
Comment 1: I noted that an institutional ethics committee has not endorsed the study. This is necessary under such research designs but especially considering genotyping of individuals was part of the methodology.
Reply: Thank you for pointing out the need for ethical approval clarification. We included the Institutional Review Board Statement in the manuscript. In response to your comment, we have now also added this information to the Materials and Methods section to provide further clarity on the ethical endorsement of the study.
Line 76-79 (method):
This study was approved by the Ethics Committee, School of Medicine, Saga University, Saga, Japan (No. R2-44 and R3-9). Written informed consent was obtained from all participants before any study procedure was performed.
Comment 2: The supplementary data commentary on cross-reactivity to SARS-CoV-2 delta and omicron variants was not completed. Readers might like to read the commentary but also see the figure, given it adds interest to the paper.
Reply: Thank you for your valuable feedback. We have addressed the comment regarding the cross-reactivity to SARS-CoV-2 delta and omicron variants by including the relevant commentary in both the Materials and Methods section and the Results section. Additionally, we have added a figure to enhance the readers' understanding and interest in this aspect of the study.
Line 152-161 (method)
2.6. Cross-reactivity with SARS-CoV-2 Delta and Omicron variants
Assessing vaccine efficacy against cross-reactivity with SARS-CoV-2 variants is essential to ensure that vaccines continue to protect against viral changes. Samples with sufficient cell numbers for this analysis were evaluated for reactivity against two different peptide mixtures: the original and delta variants (PM-SARS2-SMUT06-1) and the original and omicron variants (PM-SARS2-SMUT08-1). Cross-reactivity was evaluated between the original and delta variants in PBMCs from 74 blood samples from subcohort 1 (three weeks after the first dose and one and three months after the second dose) and subcohort 2, and between the original and omicron variants in PBMCs from 55 samples from subcohort 1 (six months after the second dose and one month after the third dose).

Figure 3. Cross-reactivity to SARS-CoV-2 delta and omicron variants. Cross-reactivity was assessed between the original and delta variants in PBMCs from 74 blood samples from subcohort 1 (three weeks after the first dose, and one and three months after the second dose) and subcohort 2, and between the original and omicron variants in PBMCs from 55 samples from subcohort 1 (six months after the second dose and one month after the third dose). PBMC, peripheral blood mononuclear cells.
Round 2
Reviewer 1 Report
Comments and Suggestions for Authors
The authors have properly addressed all concerns raised during the review process.